# Efficacy of Tixagevimab/Cilgavimab as Pre-Exposure Prophylaxis against Infection from SARS-CoV-2 and Severe COVID-19 among Heavily Immunocompromised Patients: A Single-Center, Prospective, Real-World Study

**DOI:** 10.3390/v16081345

**Published:** 2024-08-22

**Authors:** Dimitrios Basoulis, Elpida Mastrogianni, Georgios Karamanakos, Aikaterini Gkoufa, Vasiliki E. Georgakopoulou, Sotiria Makrodimitri, Maria N. Gamaletsou, Antonios Markogiannakis, Nikolaos V. Sipsas

**Affiliations:** 1Infectious Diseases Unit, Pathophysiology Department, Medical School, National and Kapodistrian University of Athens, General Hospital of Athens Laiko, 11527 Athens, Greece; katergouf@yahoo.gr (A.G.); vaso_georgakopoulou@hotmail.com (V.E.G.); sotiriamakrod@yahoo.gr (S.M.); magama@med.uoa.gr (M.N.G.);; 2Emergency Department, General Hospital of Athens Laiko, 11527 Athens, Greece; elpidamastrogianni@gmail.com; 3First Propaedeutic Internal Medicine Department, General Hospital of Athens Laiko, 11527 Athens, Greece; gkaramanakos@gmail.com; 4Pharmacy, General Hospital of Athens Laiko, 11527 Athens, Greece; mark@laiko.gr

**Keywords:** COVID-19, monoclonal antibodies, tixagevimab/cilgavimab, immunocompromised patients

## Abstract

Background: COVID-19 continues to pose a threat to immunocompromised individuals, even with vaccination. The monoclonal antibodies (mAbs) tixagevimab/cilgavimab (TXG/CIL) provide targeted prophylaxis against SARS-CoV-2 with the benefit of a prolonged half-life. Although approved for COVID-19 prevention, there is limited data on their effectiveness among heavily immunocompromised populations. Methods: We conducted a prospective, observational study at Laiko General Hospital, Athens, Greece, from August to December 2022 to investigate the efficacy of TXG/CIL as a form of pre-exposure prophylaxis in immunocompromised patients. Data on breakthrough SARS-CoV-2 infections were collected over a six-month follow-up period. Results: Of the 375 participants (mean age 61.3 ± 14.1 years; 59.7% male), 76 (20.3%) developed breakthrough SARS-CoV-2 infections, with an incidence of 3.81 cases/100 patient months. Hospitalization was required for 21 patients (5.6%), with a median stay of 14 days. Seven deaths were recorded, with only one attributed to COVID-19. Previous infection (OR 0.46, 95% CI 0.26–0.82) and hybrid immunity (OR 0.52, 95% CI 0.29–0.92) can protect against new infection. Solid organ malignancy significantly increased the risk of severe outcomes among those infected (OR 7.4, 95% CI 2.2–24.7, *p* = 0.001). Conclusions: TXG/CIL provides effective prophylaxis against COVID-19 in immunocompromised patients. Future strategies should focus on developing new mAb combinations to address emerging SARS-CoV-2 variants and protect vulnerable populations.

## 1. Introduction

Despite being appropriately vaccinated, immunocompromised individuals still face elevated risks of breakthrough SARS-CoV-2 infections and severe clinical outcomes from COVID-19 compared to the general population [1]. Monoclonal antibodies are generated in vitro to target specific bacteria or viruses. After being injected into the body, these antibodies offer immediate protection against infection by identifying and attaching to the targeted pathogen [2]. The development of protective monoclonal antibodies against SARS-CoV-2, used as pre-exposure prophylaxis, has been a valuable addition to the armamentarium against the pandemic.

Tixagevimab/cilgavimab (TXG/CIL) is a monoclonal antibody combination aimed at the RBD domain of the spike protein of the SARS-CoV-2 virus [3]. This combination, compared to other monoclonal antibodies, has the added benefit of a prolonged half-life, making it an ideal candidate for use in pre-exposure prophylaxis. In the PROVENT trial, this combination showed a relative risk reduction of 76.7% (95% CI 46–90) for symptomatic COVID-19 infection [4]. Yet only 3.2% of the study population were immunocompromised. Most of the recipients of monoclonal antibodies were identified as high-risk patients for COVID-19 due to cardiometabolic conditions. TXG/CIL received approval for the prevention of COVID-19, but there was a lack of data on immunocompromised individuals who would likely benefit the most.

Real-world studies comprising immunocompromised individuals are scarce. One of the largest cohorts, reported from France, with 1112 participants and a median follow-up of 2 months, showed a rate of 49/1112 (4.4%) of confirmed COVID-19 cases among TXG/CIL recipients [5]. A meta-analysis of TXG/CIL studies showed a significant reduction in COVID-19 cases among antibody recipients compared to those not receiving TXG/CIL [6]. Overall, TXG/CIL can reduce the risk of SARS-CoV-2 infection by 70%.

It needs to be noted that TXG/CIL has been approved for use against the Delta strain, but it is currently not authorized for use by either the FDA or EMA since newer viral strains during the Omicron era have displayed significant resistance to neutralization. XB1.5 strains have been calculated to have a greater than 5000-fold resistance, and BQ1.1 greater than 2000-fold resistance [3].

The aim of our study is to describe our experience in Greece regarding the efficacy of TXG/CIL as a form of pre-exposure prophylaxis for COVID-19 in a real-world population of heavily immunocompromised patients.

## 2. Materials and Methods

This is a prospective, observational, non-interventional study conducted at Laiko General Hospital, Athens, Greece, which is a 580-bed teaching hospital. Laiko Hospital serves mainly immunocompromised patients, especially patients with hematological malignancies. There is a hematopoietic stem cell transplantation (HSCT) unit, a solid organ (liver and kidney) transplantation program, the oncology unit, as well as units for patients with autoimmune rheumatological diseases receiving immunosuppressive and immunomodulatory treatments. TXG/CIL became available for use in Greece on 28th July 2022. The high-risk groups of patients suitable for pre-exposure prophylaxis were determined by the Greek Ministry of Health and comprised mainly heavily immunocompromised patients [7]. TXG/CIL was approved as a single 150/150 dose initially and was prioritized for immunocompromised patients. During the end of our study period and foreseeing the emergence of resistant strains based on international guidelines, patients began receiving two doses of TXG/CIL (28 participants). Our hospital was one of the few in Athens designated to administer monoclonal antibodies, not only to our patients but also to patients referred to us from other hospitals or physicians in the greater area of Athens. The program spanned five months, from 1 August 2022 to 31 December 2022.

With regards to predominant strains, from the study’s start to February 2023, the predominant strain in Greece was the B2.75 lineage, against which TXG/CIL maintained activity. From March 2023 onwards, XBB1.5 gained predominance. During the study period and the follow-up, COVID-19 hospital admissions were lower than in 2022, ranging between 1000 and 1500/week (with the peak during December and January). Upper respiratory tract infection hospital visits increased compared to the previous year, ranging from roughly 25 to 125 infections per 1000 emergency department visits. Test positivity was steady during this period, with about 10% of tests performed [8].

All consecutive patients who received at least one dose of TXG/CIL during the study period were included in the study. The study protocol was approved by the Ethics Committee of the Laiko Hospital. Demographic and medical history data were collected during the injection appointment using a structured data form. Information related to breakthrough SARS-CoV-2 infection and the development of severe COVID-19 requiring hospitalization was obtained prospectively from physicians’ electronic health records over the 6-month follow-up period. COVID-19 infections were determined based on positive SARS-CoV-2 PCR results. Indications for hospital admission and causes of death were based on treating physician information.

The primary outcome examined was breakthrough SARS-CoV-2 infection, with secondary outcomes including severe COVID-19-related hospitalizations and mortality. Patients who tested positive for COVID-19 within 5 days after receiving TXG/CIL were excluded from the analysis, as they were likely already infected at the time of the injection. Hybrid immunity was defined as having had a COVID-19 infection in the past and having received a vaccination as well.

Descriptive statistics are presented as counts (%) for categorical variables and as medians (25th, 75th percentile) for non-normally distributed continuous variables or as means ± standard deviation (SD) for normally distributed continuous variables. The normality of the distribution was examined using the Kolmogorov–Smirnov test. Group comparisons were performed using Student’s *t*-test and the Mann–Whitney U test for normally and non-normally distributed continuous variables, respectively, with the chi-square test also used for categorical variables. Results with a *p*-value < 0.05 were considered statistically significant. The analysis was performed using SPSS Statistics for Windows, Version 25.0 (2017, IBM Corp. Armonk, NY, USA).

## 3. Results

We enrolled 375 individuals with a mean age (±SD) of 61.3 ± 14.1; 224/375 (59.7) were male. Basic demographic information is presented in Table 1. Over the course of the 6-month follow-up, 76/375 (20.3%) patients tested positive for SARS-CoV-2 with a calculated incidence of 3.81 cases/100 patient months. During the follow-up period, there was a large variation in monthly incidence: 3.52, 5.1, 3.58, 5.27, 1.99, and 3.31 cases/100 patient months from month 1 to month 6, respectively. The median (IQR 25th–75th) time for the development of breakthrough SARS-CoV-2 infection after the injection of monoclonal antibodies was 2.51 (1.45–3.92) months. There was no difference in time to positivity between those receiving one dose or two doses of TXG/CIL (2.52 vs. 1.48 months, *p* = 0.763). Twenty-one patients (21/375, 5.6%) required hospitalization with a median (IQR) hospital stay of 14 (9–14) days. Over the 6-month follow-up period, we recorded 7 deaths among the study population, with 3 of them among the 76 patients with breakthrough SARS-CoV-2 infection; only 1 of these deaths was attributed to COVID-19 by the caring physicians.

Few statistical differences were identified between patients who developed breakthrough SARS-CoV-2 infection compared to those who did not (Table 1). Previous infection (OR 0.46, 95% CI 0.26–0.82) and hybrid immunity (OR 0.52, 95% CI 0.29–0.92) can protect from new infection. The last administered vaccine was at a median (IQR) time interval of 8.12 (5.12–11.2) months before TXG/CIL injection, and this time did not differ between the positive and negative groups (8.02 vs. 8.2 months, respectively, Mann–Whitney *p* = 0.614). Previous infection was recorded at a median (IQR) of 7.39 (5.39–9.94) months before injection, and this time was also not statistically different between the positive and negative groups (8.26 vs. 7.33 months, respectively, Mann–Whitney *p* = 0.629).

In Table 2, the differences between participants who developed severe COVID-19 requiring hospitalization vs. those who did not are presented. A diagnosis of solid organ malignancy significantly increased the chances of hospitalization (OR 7.4, 95% CI 2.2–24.7, *p* = 0.001), whereas previous infection or hybrid immunity (in this population, they were identical) was not protective (OR 0.11, 95% CI 0.014–0.9, *p* = 0.04).

## 4. Discussion

In our cohort of heavily immunocompromised patients (58% with active hematological malignancy; 31% solid organ transplant recipients), 76/375 (20.3%) of recipients of pre-exposure prophylaxis with TXG/CIL developed breakthrough SARS-CoV-2 infection over the 6-month period of follow-up. Twenty-one patients (21/375, 5.6%) developed severe COVID-19 requiring hospitalization, while only three deaths were recorded among TXG/CIL recipients with breakthrough SARS-CoV-2 infection. Only one death was attributed to COVID-19.

Compared to the published literature, the rate of breakthrough SARS-CoV-2 infections is significantly higher. With regards to specific immunocompromising conditions in particular, in an Italian study amongst hematological patients using an inverse probability of treatment weighting analysis, researchers found that 16.7% of participants in the treatment arm had breakthrough infections compared to 24.8% in the control arm [9]. In our own study, we found that 50 out of 219 (22.8%) hematological patients had breakthrough infections. In a previous study in Greece that focused on multiple myeloma patients, only 8.1% experienced breakthrough infections [10]. Regarding kidney transplant recipients, a study in Bari, Italy, reported that 13.3% experienced breakthrough infections compared to our findings of 17.7% (21 out of 118) [11].

There are several reasons for these differences. Firstly, 92.5% of the participants only received a single dose of TXG/CIL. In our study, we did not observe a difference between those who received one or two doses, but due to the small sample size, it may have been impossible to detect such a difference. This difference is important, as previous research has extensively documented a difference in COVID-19 incidence based on the TXG/CIL dosing regimen [6]. Secondly, since January 2023, there has been a consistent increase in the representation of the XBB 1.5 strain in Greece, which has been found to undergo resistance to neutralization using TXG/CIL [12,13]. Thirdly, in contrast to the PROVENT study, our study population comprised entirely immunocompromised individuals; therefore, it is reasonable to anticipate reduced protection.

An interesting finding in our study was that protection from future infection seemed to not be affected by previous immunization. Since a large proportion of our participants were B-cell-depleted patients, it is likely that vaccine effectiveness was compromised [14]. On the other hand, either hybrid immunity or previous infections were found to be protective. This is consistent with the published literature showing that previous COVID-19 infection has been linked to improved vaccine immunogenicity [15,16]. Finally, we observed a seven-fold increase in the risk of hospitalization among patients with solid tumors and breakthrough SARS-CoV-2 infection. According to the published literature, having an active solid tumor is a significant poor prognostic factor [17]. In contrast, hematological malignancy did not increase the risk of hospitalization among patients with breakthrough SARS-CoV-2 infection. Although we do not have available treatment data, we believe that an increased use of immunomodulatory monoclonal antibody treatments among patients with solid tumors may explain the heightened risk of hospitalization. On one hand, PD-1 and CTLA4 antibodies have been shown to lead to an increase in COVID-19 infections post-immunization, but on the other hand, they have also been associated with better outcomes [18,19]. Other studies have shown opposing results with improved vaccine serological responses to PD-1/PD-L1 blockade recipients [20].

The data deriving from a single center, even though patients were referred to us, limits the generalizability of our study. Additionally, the lack of data regarding viral lineages in those infected needs to be considered when interpreting results. Our study’s main strength lies in the selection of heavily immunocompromised patients.

## 5. Conclusions

In conclusion, we found that TXG/CIL seems to be an effective preventative measure for immunocompromised individuals. Further studies are required with newer monoclonal antibody combinations and control groups in real-world settings to further solidify these findings. Among the recipients of TXG/CIL, hybrid immunity and/or vaccination seem to be further protective.

## Figures and Tables

**Table 1 viruses-16-01345-t001:** Basic characteristics of study population and comparison between those infected and healthy subjects.

	Total N = 375	SARS-C0V-2 Negative N = 299	SARS-C0V-2 Positive N = 76	*p*
Age	61.3 ± 14.1	61.7 ± 13.8	58.3 ± 16.1	0.091
Male gender	224 (59.7)	177 (59.2)	47 (61.8)	0.697
Coronary heart disease	46 (12.3)	37 (12.4)	9 (11.8)	1
Heart failure	30 (8)	20 (6.7)	10 (13.2)	0.093
COPD/asthma	51 (13.6)	40 (13.4)	11 (14.5)	0.851
Diabetes mellitus	87 (23.2)	69 (23.1)	18 (23.7)	0.880
Chronic kidney disease	117 (31.2)	96 (32.1)	21 (27.6)	0.491
Autoimmune disease	29 (7.7)	24 (8)	5 (6.6)	0.813
Solid organ malignancy	76 (20.3)	60 (21.1)	16 (21.1)	0.873
Hematological malignancy	219 (58.4)	169 (56.5)	50 (65.8)	0.154
HSCT recipient	11 (2.9)	7 (2.3)	4 (5.3)	0.244
Solid organ transplant recipient	118 (31.5)	91 (32.4)	21 (27.6)	0.490
SARS-CoV-2 vaccination				0.490
Unvaccinated	10 (2.7)	9 (3)	1 (1.3)	
One dose	8 (2.1)	6 (2)	2 (2.6)	
Two doses	29 (7.3)	20 (6.7)	9 (11.6)	
Three doses	142 (38.1)	110 (36.8)	32 (42.1)	
Four doses	176 (46.9)	146 (48.8)	30 (39.5)	
Five doses	10 (2.7)	8 (2.7)	2 (2.6)	
TXG/CIL dose				0.230
150 mg/150 mg	347 (92.5)	274 (91.6)	73 (96.1)	
300 mg/300 mg	28 (7.5)	25 (8.4)	3 (3.9)	
Previous COVID-19 infection	138 (36.8)	120 (40.1)	18 (23.7)	**0.008**
Hybrid immunity	130 (34.7)	112 (37.5)	18 (23.7)	**0.03**

*p* < 0.05 denotes statistical significance and appears in bold. COPD: chronic obstructive pulmonary disease, HSCT: hematopoietic stem cell transplantation, and TXG/CIL: tixagevimab/cilgavimab.

**Table 2 viruses-16-01345-t002:** Comparison between participants requiring hospitalization and those who did not.

	Not Hospitalized N = 55	Hospitalized N = 21	*p*
Age	58.4 ± 16.5	60.1 ± 14.6	0.696
Male gender	35 (63.6)	12 (57.1)	0.609
Coronary heart disease	7 (12.7)	2 (9.5)	1
Heart failure	9 (16.4)	1 (4.8)	0.268
COPD/asthma	8 (14.5)	3 (14.3)	1
Diabetes mellitus	15 (27.3)	3 (14.3)	0.366
Chronic kidney disease	16 (29.1)	5 (23.8)	0.778
Autoimmune disease	2 (3.6)	3 (14.3)	0.126
Solid organ malignancy	6 (10.9)	10 (47.6)	**0.001**
Hematological malignancy	34 (61.8)	16 (76.2)	0.288
HSCT recipient	4 (7.3)	0 (0)	0.571
Solid organ transplant recipient	18 (32.7)	3 (14.3)	0.153
SARS-CoV-2 vaccination	0.673		0.213
Unvaccinated	1 (1.8)	0 (0)	
One dose	2 (3.6)	1 (4.8)	
Two doses	7 (12.7)	2 (9.5)	
Three doses	20 (36.4)	5 (23.8)	
Four doses	16 (29.1)	7 (33.3)	
Five doses	2 (3.6)	3 (14.3)	
TXG/CIL dose			1
150 mg/150 mg	53 (96.4)	20 (95.2)	
300 mg/300 mg	2 (3.6)	1 (4.8)	
Previous COVID-19 infection	17 (30.9)	1 (4.8)	**0.017**
Hybrid immunity	17 (30.9)	1 (4.8)	**0.017**

*p* < 0.05 denotes statistical significance and appears in bold. COPD: chronic obstructive pulmonary disease, HSCT: hematopoietic stem cell transplantation, and TXG/CIL: tixagevimab/cilgavimab.

## Data Availability

Data will be made available at the Pergamos repository of the National and Kapodistrian University of Athens upon acceptance.

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
