# Peer review of "Efficacy of Tixagevimab/Cilgavimab as Pre-Exposure Prophylaxis against Infection from SARS-CoV-2 and Severe COVID-19 among Heavily Immunocompromised Patients: A Single-Center, Prospective, Real-World Study"

_viruses, 2024, doi:10.3390/v16081345_

Round 1

Reviewer 1 Report

Comments and Suggestions for Authors

Authors present a large series of heterogenous patients to try and define the efficacy of TXG/CIL for pre-exposure prophylaxis against COVID-19. My comments and suggestions are below:

1. In the introduction- please make it clear that TIG/CIL has activity against certain strains of virus and is currently not approved for use. You may also provide data regarding he duration if t1/2 and PK characteristics. This is important to define the exclusion of patients who tested positive within 5 days of administration, while the peak efficacy maybe much later.

2. While COVID- related hospitalizations are listed as the secondary outcome, data regarding them is scarce.

- definition of infection

- how were other causes ruled out

- who arbitrated for any discripancies

- among these IC patient- there can be multiple reasons for hospitalization. so this distinction has be clear.

- clinical characteristics of patients hospitalized? how sick? how many received supplemental O2, mechanical ventilation, treatment, complications etc.

- definition of hybrid immunity? how was it measured? definition?

- time from last administration of TXG/CIL to COVID diagnosis between lower and higher dose ?

- were any of these patients undergo escalation of IS, treatment for rejection, increased intensity of IS, plasmapheresis or B-cell depletion?

- what was the time from transplant? if recent- what induction therapy was used?

- solid tumors is a big class- any subgroup numbers? any unique new treatments including immunotherapy?

3. A context would be helpful for historical purposes:

- what was/were the current predominant strains during the study period?

- were any of the strains, especially in later part of the study, suspected to be of variety that is not protected by TXG/CIL? is there any data for this?

- what were the community prevalent rated of COVID during the study  period?

Comments on the Quality of English Language

Overall acceptable. Needs some formatting and corrections of spellings.

Author Response

Comment 1: In the introduction- please make it clear that TIG/CIL has activity against certain strains of virus and is currently not approved for use. You may also provide data regarding he duration if t1/2 and PK characteristics. This is important to define the exclusion of patients who tested positive within 5 days of administration, while the peak efficacy maybe much later.

Response 1: We added this paragraph "It needs be noted that TXG/CIL has been approved for use against Delta strain, but it is currently not authorized for use by either FDA or EMA, since newer viral strains during the Omicron era have displayed significant resistance to neutralization. XB1.5 strains have been calculated to have a greater than 5000-fold resistance and BQ1.1 greater than 2000-fold resistance." With regards to the PK data and halflife, albeit we agre it would be useful information, we feel that the methods section is already too long for a brief report and it would burden the manuscript more at this point. 

Comment 2: While COVID- related hospitalizations are listed as the secondary outcome, data regarding them is scarce.

Response 2: We have added these lines

"COVID-19 infections were determined based on positive SARS-CoV-2 PCR results. Indication for hospital admission and causes of death were based on treating physician information." With regards to infection, given that it was based on PCR data the information is clear. We acknowledge that information provided by patients and physicians with regards to reasons for hospitalization and/or death is dependent on their honesty but it was not something that we would have been able to verify in this study. We do not have details regarding the severity of COVID-19 infections aside from need to be hospitalized or not.

We also added this line "Hybrid immunity was defined as having had a COVID-19 infection in the past and received vaccination as well."

Patients receiving the high dose and consequently contracting COVID-19 are only three, but we have added a line for this in the results. "There was no difference in time to positivity between those receiving one dose or two doses of TXG/CIL (2.52 vs 1.48 months, p=0.763). "

We do not have information with regards to cancer treatments, immunosuppression changes, transplant treatments, time from transplant, transplant rejection

Comment 3: A context would be helpful for historical purposes

Response 3: We have added this in methods 

With regards to predominant strains, from study start and up until February 2023, the predominant strain in Greece was of the B2.75 lineage, against which TXG/CIL maintained activity. From March 2023 and onwards, XBB1.5 has gained predominance. During the study period and the follow-up, COVID-19 hospital admissions were lower than in 2022, ranging between 1000-1500/week (peak during December and January). Upper respiratory tract infection hospital visits were increased compared to the previous year, ranging from roughly 25 to 125 infections per 1000 emergency department visits. Test positivity has been steady during this period at about 10% of tests performed." COVID-19 specific prevalence information was not provided by the referenced national public health organisation surveillance reports.

Reviewer 2 Report

Comments and Suggestions for Authors

The study is both intriguing and of significant clinical relevance. I have the following comments:

1.      Based on the results presented in the tables, it appears that the primary protective factor against both infection and hospitalization is previous infection or hybrid immunity. How do these data support the conclusion regarding the protective role of TXG/CIL administration?

2.      In the discussion, the authors suggest that the worse outcomes observed in patients with solid tumors might be due to the use of immunomodulatory antibodies in cancer treatment. However, in most cases, these treatments involve anti-PD1 or anti-CTLA4 antibodies, which enhance adaptive immunity. There are also reports indicating an increased response to vaccines in patients undergoing such therapies.

Comments on the Quality of English Language

There are only some type errors

Author Response

Comment 1: Based on the results presented in the tables, it appears that the primary protective factor against both infection and hospitalization is previous infection or hybrid immunity. How do these data support the conclusion regarding the protective role of TXG/CIL administration?

Response 1: In order to make an assessment on TXG/CIL compared to people not receiving the combination we would need a control group, which we do not have. Our data shows an overall low rate of COVID-19 infections (though as mentioned still greater than the rate descirbed in the sutdies leading to TXG/CIL approval) and an even lower rate of hospitalizations or deaths. Within this group of people who all received prophylactic antibodies, hybird immunity was further protective. We understand your concern that without a control group, our statement might be too bold and we have made a small change to the conclusions paragraph.

Comment 2: In the discussion, the authors suggest that the worse outcomes observed in patients with solid tumors might be due to the use of immunomodulatory antibodies in cancer treatment. However, in most cases, these treatments involve anti-PD1 or anti-CTLA4 antibodies, which enhance adaptive immunity. There are also reports indicating an increased response to vaccines in patients undergoing such therapies.

Response 2: The interplay of these molecules and COVID-19 is complicated. There have been reports of increased COVID-19 cases post-immunization in this group on one hand and on the other hand reports that PD-1 and CTLA4 blockade are actually favourable since overexpression leads to worse outcomes. We have added some further comments and citations.

"On  one hand, PD-1 and CTLA4 antibodies have been shown to lead to an increase in COVID-19 infections post immunization but on the other hand they have also been associated with better outcomes. [18, 19] Other studies have shown opposing results with improved vaccine serological responses to PD-1/PD-L1 blockade recipients. [20]"

Reviewer 3 Report

Comments and Suggestions for Authors

This is an important study about a Greek population of 375 immunocompromised patients receiving a one-time dose of Tixagevimab/Cilgavimab and then followed to assess breakthrough infections with SARS-CoV-2 to develop COVID-19.  To date, other studies have not had exclusively immunocompromised patients, but a mixture of immunocompromised and non-immunocompromised patients.  One advantage is that it focuses on solid and non-solid cancer patients (e.g. lymphoma and other blood cancers) and breakthrough infections. An interesting finding was that previous immunization did not protect against future infection, possibly due to new strains circulating. Another interesting finding is that previous COVID-19 infection with and without immunization (hybrid immunity) protected against further infections and hospitalization better than immunization. Interestingly, solid tumor patients had a higher incidence of hospitalizations with COVID-19 than other immunocompromised patients.

Major Critiques:

Please include a description of the type of statistical analyses used in the study in the Materials and Methods.  Without this, the study cannot be evaluated properly. SPSS is the program that implements the statistical testing, not a description of the tests used.

Minor Critiques:

Please detail the dosing in Materials and Methods.

A table of exclusions should be included for clarity.

Author Response

Comment 1: We have added a paragraph explaining statistical methods

Comment 2: Please detail the dosing in Materials and Methods. A table of exclusions should be included for clarity.

 Response 2: We added a line for the dosing of TXG/CIL in methods "During the end of our study period and foreseeing the emergence of resistant strains and based on international guidelines, patients began receiving two doses of TXG/CIL (28 participants)."

Since patients were referred to us for TXG/CIL we did not apply any exclusion criteria to these referrals. We are not aware if referring physicians applied any.

Round 2

Reviewer 2 Report

Comments and Suggestions for Authors

The authors address both of my concerns by commenting on the role of immunotherapy and acknowledging a limitation of the study in the conclusion.